# Misdiagnosis of Acute Limb Ischemia from Non-Vascular Specialists Results in a Delayed Presentation and Negatively Affects Patients’ Outcomes

**DOI:** 10.3390/medsci13010021

**Published:** 2025-02-20

**Authors:** Michalis Pesmatzoglou, Stella Lioudaki, Nikolaos Kontopodis, Ifigeneia Tzartzalou, Konstantinos Litinas, George Tzouliadakis, Christos V. Ioannou

**Affiliations:** Vascular Surgery Unit, Department of Vascular and Cardiothoracic Surgery, Medical School, University of Crete, 71500 Crete, Greece; michalis.pesmatzoglou@gmail.com (M.P.); lioudakistella@hotmail.com (S.L.); tzartzalouif@gmail.com (I.T.); litinask@gmail.com (K.L.); georgios.93tz@gmail.com (G.T.); ioannou@med.uoc.gr (C.V.I.)

**Keywords:** embolism, thrombosis, diagnosis delayed, arterial disease peripheral

## Abstract

Background/Objectives: Acute Limb Ischemia (ALI) is a vascular emergency which is accompanied by a significant risk of limb loss or even death. Rapid restoration of arterial perfusion using surgical and/or endovascular techniques is crucial for limb salvage. Undeniably, an accurate and prompt diagnosis is the first step to improve patient prognosis. The typical clinical presentation is not always present and the variety of symptoms may result in non-vascular specialists missing the diagnosis. Methods: In this single-center retrospective descriptive study, we reviewed all patients hospitalized between January 2018 and January 2024 for ALI. Patients who were initially misdiagnosed, causing a delayed diagnosis > 24 h, and who therefore did not receive timely treatment, were identified. Moreover, patients with a timely diagnosis of ALI who were treated in our institution during the same time period were collected. Results: Among 280 ALI patients, 14 were initially misdiagnosed. The median time from initial symptoms to definite diagnosis was 38.8 days (range 1.5–365). Several specialties such as orthopedic surgeons, neurologists, and general practitioners were involved in patients’ initial assessment. Three patients underwent primary amputation due to irreversible ALI, while nine underwent revascularization and one conservative treatment. Thirty-day limb salvage rate was 9/14 and thirty-day mortality was observed in one patient. Secondary interventions were needed in 65% of these cases. Patients with a delayed ALI diagnosis, when compared to those with a timely diagnosis, presented a significantly lower limb salvage rate (65% vs. 89%, *p*-value = 0.02) and a significantly higher rate of reinterventions (65% vs. 18%, *p*-value < 0.001). Conclusions: Many patients with ALI are primarily referred to non-vascular specialties. Misdiagnosed and mistreated ALI negatively affects outcomes.

## 1. Introduction

Acute Limb Ischemia (ALI) is a surgical emergency that poses a risk to limb viability and in extreme cases to patients’ lives [1]. A prompt and accurate diagnosis is of paramount importance in order to proceed with the appropriate therapeutic plan/management and restore arterial limb perfusion in a timely manner, in order to avoid irreversible ischemia and limb loss.

According to the severity of symptoms, and the viability of the limb, ALI is classified in four categories, namely viable, marginally threatened, immediately threatened, and finally irreversible ischemia [2]. Taking into account the classification of ALI, an appropriate treatment plan should be initiated. In the presence of a viable limb, anticoagulant treatment is administered and revascularization is performed in an urgent setting. If the limb is threatened, immediate and emergent restoration of limb perfusion is necessary with either open surgical or endovascular methods such as percutaneous mechanical thrombectomy, catheter-directed thrombolysis, and/or thrombectomy/embolectomy and peripheral bypass. When irreversible ischemia has been established, revascularization is not appropriate and amputation should be performed [3,4,5]. Despite all of the major advances in treatment, acute limb ischemia is a high-mortality condition with high rates of limb loss and mortality [2,6].

Remarkably, the clinical presentation of ALI may be variable and the typical symptomatology may be absent in a significant proportion of patients [7]. Taking into account that vascular surgeons are not usually the first medical specialty that evaluates these patients (vascular surgeons mostly staff departments in tertiary care hospitals, while they have no role in primary care centers), it is not strange that the correct diagnosis may be initially missed by other first-line, non-vascular specialties [8]. Indeed, anecdotal experience suggests that it is common for patients with lower limb pain and related symptoms to be initially evaluated by other medical specialties such as general practitioners, orthopedics, etc., which is especially true for patients without the typical atherosclerotic risk factors. Unfortunately, although this seems a reasonable argument, there are no published data to support such a claim. Moreover, the acute presentation of this condition, along with the limited resistance of neurologic and muscular tissue to ischemia and the significant risk of limb loss that previous research has indicated after ALI, render timely diagnosis and treatment critical for a favorable patient outcome [6,7,8]. Remarkably, the rate of ALI patients who are initially misdiagnosed and how this affects patients’ outcomes are poorly reported in the literature. With the current study, we aim to record cases with ALI that were initially examined by non-vascular specialties and misdiagnosed, in order to evaluate the delay of referral to a vascular specialist and the impact that this had on the prognosis of the patients.

## 2. Materials and Methods

### 2.1. Study Design and Study Population

This is a single-center, retrospective descriptive observational study, which recorded all patients hospitalized in our department between January 2018 and January 2024 with ALI. From the hospital’s electronic database, all patients that were assigned an ICD-10 code I74, I74.1, I74.2, I74.3, I74.4, I74.5, I74.6, I74.7, I74.8, or I74.9 were identified and their medical records were retrieved. ALI was defined according to international guidelines as a sudden decrease in arterial perfusion of the limb, with a potential threat to the limb survival, requiring urgent evaluation and management [7]. From the total pool of patients, we identified those who were at first misdiagnosed by non-vascular specialists, and who therefore did not receive proper management and came to the Emergency Department (ED) at a later time, with various degrees of advanced limb ischemia. Patients who were initially evaluated by other medical specialties, assigned a diagnosis different than ALI, and were diagnosed with ALI > 24 h after initial presentation were included in the analysis and comprised the study group. Patients with a timely diagnosis and standard treatment comprised the control group. All patients that were misdiagnosed were initially examined in other institutions or in private practice, by non-vascular specialties. Definitive diagnosis of ALI was made by vascular surgeons at the emergency department of the institution conducting this study, which provides the only vascular service in a large insular area in Greece. Therapeutic management and interventional/surgical procedures were performed at the same institution. The diagnosis of ALI was mainly based on the history and clinical examination of patients. This included palpation of pulses in all four limbs, detection of arterial and venous flow at the level of the malleolus with a hand-held Doppler device, and clinical evaluation of sensory and motor function. Patients diagnosed with ALI were subsequently subjected to CT angiography in order to determine the appropriate therapeutic approach.

The variables that were collected regarded delay time (days from first medical consultation to final diagnosis), the medical specialists that were involved in the diagnostic process, subsequent etiology of the ischemia, severity of ischemia on presentation to the ED, and the treatment modality of choice. The primary outcome of the analysis was limb salvage rate, while secondary outcomes included 30-day mortality, ankle–brachial index at discharge, need for reinterventions, ICU stay, and hospital stay. This information was detected through the medical records of the patients and telephone contact where there were incomplete and missing data. The current study has been reported following the STROBE guidelines for reporting observational studies (http://www.strobe-statement.org, accessed on 12 September 2024). The institutional review board has approved the conduction of the present study. Study participants provided informed consent.

### 2.2. Statistical Analysis

Continuous variables are reported as median and range. Categorical variables are presented as count and percentage. Comparisons regarding continuous variables were made with the non-parametric Mann–Whitney test, while categorical variables were compared using the chi-square test. Statistical significance was defined as a *p*-value < 0.05 and all statistical analyses were performed using the Statistical Package for the Social Sciences v21.0 (IBM Inc., Chicago, IL, USA) software.

## 3. Results

### 3.1. Study Population

During the study period, 280 patients were treated for ALI in our institution. There were 205 male and 75 female patients, with a median age of 72 years (33 years–101 years). Most patients complained of sudden-onset lower limb acute pain accompanied by numbness (n = 203 patients, 72% of the study population), while the remaining cases reported acute deterioration of walking ability with intermittent claudication presenting after walking only a few meters (n = 77 patients, 28% of the study population). These were mostly patients with an existing diagnosis of peripheral arterial disease. In a subgroup representing 9% (n = 25 patients) of cases, previous revascularization procedures were documented. Regarding the etiology of ALI, embolism was recorded in the majority of cases (n = 146 patients, 53% of the study population) and thrombosis was the second most common cause (n = 92 patients, 33% of the study population), while symptomatic popliteal artery aneurysms (n = 8), trauma (n = 8), and cystic adventitial disease (n = 1) were also recorded in a minority of cases. Most limbs were classified as threatened (either marginally or immediately, n = 203, 72% of cases), while viable limbs were encountered in 44 (16%) and irreversible ischemia in 33 (12%) cases. Most patients (243/280, n = 87%) were receiving either anticoagulant or antiplatelet treatment.

Fourteen patients (5% of the total population) fulfilled the above-mentioned criteria of misdiagnosed ALI and were included in the analysis. Among them, there were eight women and six men, with a mean age of 56 years old (33 yo–91 yo) and an interval between first medical consultation and final diagnosis of 38.8 days (1.5 d–365 d). Among the medical specialties who initially evaluated the patients, there were orthopedic surgeons (six cases), general practitioners (four cases), internal medicine doctors (three cases), neurosurgeons (two cases), and neurologists (one case), while in 3/14 cases more than one specialty were involved.

The most common cause of ischemia among patients with a delayed diagnosis was distal embolization (seven cases), followed by thrombosed popliteal aneurysm (three cases), thrombosis of previous atherosclerotic plaques (two cases), cystic adventitial disease of the popliteal artery (one case), and trauma (one case).

Regarding the severity of ALI during initial presentation at the Emergency Department, ten patients presented with immediately threatened limb ischemia (IIb Rutherford classification), three patients with irreversible ischemia (III Rutherford classification), and one patient with a viable limb (I Rutherford classification) [9].

### 3.2. Surgical/Endovascular Procedures

Most often, open surgical revascularization was selected as the treatment of choice (five cases), followed by endovascular mechanical thrombectomy (AngioJet Peripheral Thrombectomy system—Boston Scientific, Natick, MA, USA) combined with percutaneous transluminal angioplasty (PTA) (four cases), major amputation (three cases), open surgical thrombectomy using a Fogarty catheter (one case), and conservative treatment (one case). Among the open revascularization procedures, femoropopliteal bypass (two cases), axillobifemoral bypass (two cases), and femoro-posterior tibial bypass (one case) were included. In the PTA-treated group, stent grafts were also deployed in two cases.

### 3.3. Primary and Secondary Outcomes

At the time of the discharge, four patients had uncomplicated post-op hospitalization, while the rest had to undergo further reoperation, consisting of fasciotomy due to compartment syndrome (three cases), minor (one case) or major (two cases) amputation, revision of amputation site (two cases), and femoropopliteal bypass (two cases).

Overall, four patients had to stay in the intensive care unit (ICU), for a median duration of 5 days (2–8 days), and the median hospital stay of the patients was 10.5 days (2–45 days).

In total, limb salvage was achieved in 9/14 patients. These patients presented a mean ABI of 0.9 at the time of discharge. The 30-day mortality was around 7% (1/14 cases). Specifically, Pt #3 developed acute renal injury due to prolonged high values of creatinine phosphokinase, which was further complicated by acute respiratory distress due to the patient being bedridden, and he died during his 6th day at the ICU due to cardiopulmonary failure, caused by multi-organ failure.

During a mean follow-up of 12 months, no further mortality was recorded and no further amputation was required. Demographic and clinical details of patients are summarized in Table 1.

### 3.4. Comparative Analysis of Patients with a Delayed vs. A Timely Diagnosis

Limb salvage was achieved in 237/266 patients with a timely diagnosis, which was significantly better than that among patients with a delayed diagnosis (limb salvage 89% vs. 65%, *p*-value = 0.02). Thirty-day mortality was higher among patients in the control group (11%, 29/266 patients), but the difference was not significantly different compared to the study group (*p*-value = 0.68). Peri-procedural secondary interventions were significantly less common among patients with a timely diagnosis (48/266–18% of patients in the control vs. 9/14–65% of patients in the study group, *p*-value < 0.001). Duration of hospital stay was also significantly reduced in the control group (median 7 vs. 10.5 days, *p*-value = 0.024).

## 4. Discussion

In the current study, the primary endpoint, which was limb loss, was recorded at a rate as high as 36%, which compares unfavorably with most published reports and with our cohort of patients with a timely diagnosis. Typically, short-term limb loss is reported in around 10% of patients presenting with ALI [10]. A recent European registry recorded an amputation-free survival of 86% within 30 days, among 705 patients [5]. A previous report from the Medicare population, including almost 100,000 patients, indicated an amputation rate of 6.4% during the index hospitalization and 8.1% within 30 days of the initial procedure, which again is remarkably lower than values recorded in our study population involving patients with ALI and late presentation due to misdiagnosis [11].

On the other hand, in-hospital mortality in the current study was around 7%, which is mostly similar to typical values reported in the literature [10]. Short-term mortality after ALI is usually due to multi-organ failure, which was also the case in the single death recorded in the present report. It seems that delay in diagnosis and treatment of ALI mainly affects the fate of the limb, resulting in an excess rate of major amputations, while mortality rate may remain unaffected. Of course, the small study population of this report does not allow for a definite conclusion regarding the mortality rate of such patients. In our study population, the short-term mortality rate was higher in patients with a timely diagnosis, although this difference did not reach statistical significance. The fact that patients in the control group were older, with the typical comorbidities of patients with cardiac arrythmias or and/or peripheral arterial disease, may have contributed to this observation.

Moreover, we recorded an exceptionally high reintervention rate among patients with a delayed diagnosis. Specifically, around 65% of patients required a secondary intervention. Remarkably, two patients undergoing endovascular thrombectomy subsequently needed a secondary bypass procedure, which may be due to the fact that suboptimal results could be obtained after mechanical thrombectomy of subacute and/or chronic thrombus, resulting in re-occlusion of the vessel. Moreover, the need for revision in patients with irreversible ischemia and primary amputation was very common. It may be that BKA presented a reduced healing rate among patients with ALI and delayed presentation, although it is not clear if systematic or limb-related variables affect the healing process at a more distal level. Previous reports suggest a lower need for secondary interventions, around 20%, while others have reported rates as low as 3% [12].

Additionally, the need for ICU was encountered in >30% of patients (4/14 cases), while median length of stay was around 14 days. Possibly, patients subjected to revascularization procedures after a prolonged period of ischemia present increased need for intensive care or hospitalization due to complications such as reperfusion injury, increased complexity of primary operations, and/or need for reinterventions. It is not surprising that there were worse outcomes in patients included in the current analysis compared to the typical patient population presenting with ALI, taking into account the delayed presentation of these cases.

Common causes of ALI include thromboembolism, native plaque thrombosis, aneurysm, and thrombosis after previous revascularization procedures. Other causes such as dissection, vasculitis, popliteal entrapment syndrome, popliteal artery cystic adventitial disease, and thrombophilia are rarely encountered [9]. In our case series, embolism was the major cause of the acute ischemia, followed by thrombosed popliteal artery aneurysm, thrombosis of atherosclerotic plaque, cystic adventitial disease of the popliteal artery, and trauma. Contrary to what was actually observed, and taking into account that patients included in the current study had a late presentation, one would expect most of them to have presented with native atherosclerotic plaque thrombosis. This is usually observed in subjects with previous peripheral arterial disease who have already developed collaterals which result in less severe clinical presentation and may keep the limb viable for a considerable amount of time, as opposed to embolization, which more often results in a more severe clinical presentation. Notably, we did not record any cases of ALI after previous revascularization procedures, which has recently been reported to be an increasing cause of ALI [11]. We suspect that, in the presence of a medical history of previous vascular procedures, non-vascular specialists present a higher rate of suspicion towards vascular pathologies and are more likely to ask for a vascular surgery consultation. Therefore, misdiagnosis may not be so frequent in this setting. Representative of the delay that was observed until patient presentation is the fact that only one patient in the current series was subjected to the standard embolectomy that would be appropriate in the majority of patients in the event of a timely presentation.

A remarkable finding of this study is that 8/14 patients initially misdiagnosed were <60 years of age. It is possible that, in this young subset of patients who do not fit the typical clinical profile of patients with vascular diseases (such as old male smokers, with hypertension, dyslipidemia, and diabetes mellitus), most physicians without a high index of clinical suspicion would relate limb symptoms to other pathologies such as orthopedic or neurologic, thus missing the opportunity to evaluate arterial perfusion. Remarkably, the typical profile of patients with ALI would be elderly patients with cardiac comorbidities, those who have been already diagnosed with chronic peripheral arterial disease, or those who have been previously subjected to revascularization procedures. These patients usually present a typical medical history and course of symptoms which make it easier for non-vascular specialists to suspect ALI, in contrast to patients included in the current analysis [13,14]. Overall, patients with a delayed ALI diagnosis included in the current study are highly diverse in terms of age, causes of ALI, medical procedures performed, length of hospitalization, and other factors, but most of them do not conform to the classical profile of typical ALI cases.

As is stated at the 2020 ESVS Guidelines, many patients with ALI are primarily admitted to non-vascular specialties. In the current series, median time to definitive diagnosis of ALI by a vascular specialist was >30 days, while a recent case series recorded a median time from clinical presentation to consultation and diagnosis of around 25 h (5.1 h–79 h) [7,10]. Typically, the classic “six Ps” (pain, pallor, pulselessness, poikilothermia, paresthesia, and paralysis) make up the clinical presentation of patients with ALI. Nevertheless, all six signs are rarely encountered at once, unless there is severe ALI in a patient with otherwise normal arteries [7]. Other medical conditions that may mimic compromise of the arterial perfusion of the limb and must be excluded are deep vein thrombosis, nerve root compression, foot or ankle arthritis, popliteal artery entrapment syndrome, popliteal cystic adventitial disease, etc. [15]. In general, neurogenic claudication typically worsens during spine extension and is relieved with flexion, being associated with fatigue. Chronic iliofemoral venous thrombosis may cause venous claudication, with pain mainly located in the thigh during stressful exercise. Arthritis is accompanied by limb pain that occurs with exertion, but walking capacity varies over time, unlike typical vascular claudication [15].

Notably, in the current series, most of the limbs were classified as Rutherford class IIb (immediately threatened), while in other series delayed presentation has mostly related to viable limbs [16]. Possible explanations could be the prolonged delay in patient presentation to the ED and mismanagement by the physicians who were first called to assess the patient. In the literature, it has been indicated that this delay can be due to patient delay, delay caused by medical personnel, and waiting times in the ED and for the operating theatre [17]. However, in our case series, all patients were treated within 6 h of presentation in the ED and no further delay was noticed.

Rutherford classification is recommended for ALI, for clinical evaluation and further treatment. Grade I indicates a viable limb, Grade IIA and IIB are suggestive of a marginally and immediately threatened limb, respectively, and Grade III represents irreversible ischemia and non-viable limb [9]. In our case series, around 21% of patients presented with Rutherford grade III ischemia, which is significantly higher than the previously published data, and which affected the further treatment and overall prognosis of the patients [10].

Reperfusion injury and compartment syndrome (CS) is a common complication, especially in young patients who have had prolonged ischemia and undergone revascularization. In these patient groups, high clinical suspicion, early diagnosis and emergent treatment, or prophylactic fasciotomy can be considered [18]. In our case series, none of the patients underwent prophylactic fasciotomy, but they were all monitored closely post-op and three of them developed CS and had to undergo four-layer fasciotomy, without any further complications.

Our results indicate that there is certainly room for improvement on the diagnostic and therapeutic pathway of patients with ALI. Similarly to previous observations, it seems that primary and/or emergency department physicians may not be familiar with the clinical presentation of ALI [19]. Education of these specialties about history, symptoms, and clinical signs of this condition could contribute to increased awareness and reduce the rate of delayed diagnosis. Others have recorded that, among patients with ALI, the diagnosis was suspected in around two-thirds of cases [20]. We would agree with these authors that, if direct communication could be created between primary care facilities/emergency departments and the vascular surgeons on call at the tertiary care facility, outcomes could be improved. A target for more rapid communication between the pre-hospital and the hospital team could be an achievable goal in this setting. Timely diagnosis of ALI, in addition to the fact that it may allow for an early revascularization procedure, has further therapeutic implications. In this regard, rapid initiation of anticoagulant medication has been indicated to improve outcomes. On the other hand, delays in diagnosis and anticoagulation resulted in a significantly higher 30-day reintervention rate [21].

Limitations of the current analysis arise from the fact that collection of data was retrospective and, as such, may be prone to collection bias. Nevertheless, since an electronic search of the hospital’s database was performed which used ICD-10 codes, we are confident that all eligible patients were identified. Moreover, collection of medical records for a small medical cohort comprising our study population was straightforward.

In conclusion, patients with acute limb ischemia may not always present with typical symptoms and the diagnosis can be missed by first-line non-vascular specialists. Delayed presentation negatively affects outcomes and may result in a higher amputation rate, need for reinterventions, need for ICU, and increased hospital stay. A high degree of awareness is required from physicians performing the first evaluation to correctly identify cases with possible ALI. Development of educational activities to inform non-vascular specialists about this vascular emergency as well as establishing connection pathways between primary care physicians and vascular surgery departments may assist to reduce the rate of misdiagnosis of ALI cases.

## Figures and Tables

**Table 1 medsci-13-00021-t001:** Summary of demographic and clinical information of patients with a delayed diagnosis (ALI: Acute limb ischemia, GP: General practitioner, PA: popliteal artery, FP: femoropopliteal, AKA: Above-knee amputation, BKA: Below-knee amputation, —: none).

# Patient	Age	Delay in Presentation (Days)	Specialist Who Performed Initial Evaluation	Cause of ALI	Primary Procedure	Secondary Procedures	ICU Stay (Days)	Hospital Stay (Days)	Limb Salvage
Pt #1	67	5	GP	Thrombosed PA aneurysm	FP bypass	Fasciotomy	—	7	Yes
Pt #2	91	4	GP	Embolization	AKA	Revision	8	13	No
Pt #3	84	5	Orthopedic	Embolization	BKA	AKA	6	10	No
Pt #4	35	365	Orthopedic	Adventitial cystic disease	Angiojet and PTA	Minor amputation—transmetatarsal	—	27	Yes
Pt #5	74	90	Orthopedic	Thrombosed atheroscerotic plaque	Axillobifemoral bypass	—	4	10	Yes
Pt #6	54	1.5	GP	Thrombosed atheroscerotic plaque	Axillobifemoral bypass	—	2	11	Yes
Pt #7	33	30	GP, Internist	Thrombosed PA aneurysm	AKA	Revision	—	45	No
Pt #8	37	2	Neurosurgeon, Orthopedic	Trauma	FP bypass	Fasciotomy	—	6	Yes
Pt #9	43	2	Neurosurgeon	Embolization	Angiojet and PTA	FP bypass	—	8	Yes
Pt #10	45	3	Neurologist	Embolization	Angiojet and PTA	FP bypass	—	12	Yes
Pt #11	43	4	Neurosurgeon	Embolization	FP bypass	BKA	—	18	No
Pt #12	58	5	GP	Embolization	Angiojet and PTA	Fasciotomy	—	6	Yes
Pt #13	88	7	Internist	Embolization	Mechanical thrombectomy	—	—	20	Yes
Pt #14	67	20	Orthopedic, Internist	Thrombosed PA aneurysm	Conservative treatment	—	—	2	Yes

## Data Availability

The original contributions presented in this study are included in the article. Further inquiries can be directed to the corresponding author.

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
