# Peer review of "Misdiagnosis of Acute Limb Ischemia from Non-Vascular Specialists Results in a Delayed Presentation and Negatively Affects Patients’ Outcomes"

_medsci, 2025, doi:10.3390/medsci13010021_

Round 1

Reviewer 1 Report

Comments and Suggestions for Authors

Simple idea and hypothesis, clearly structured and well-written study. Congratulations to the authors, I enjoyed reading it 

Author Response

Comment 1: "Simple idea and hypothesis, clearly structured and well-written study. Congratulations to the authors, I enjoyed reading it".

Response 1: Thank you for your encouraging comments.

Reviewer 2 Report

Comments and Suggestions for Authors

I read an interesting paper on the evolution of patients with acute peripheral ischemia, diagnosed late.

I congratulate the authors for the honest and critical analysis of the situation, emphasizing the important role of the correct and rapid evaluation and diagnosis of this pathology.

However, I would like to suggest a few additions:

- Enriching the introductory notions regarding the current treatment of patients with acute peripheral ischemia (see the article “New Directions in the Management of Peripheral Artery Disease Chioncel, V; Brezeanu, R; Sinescu, C AMERICAN JOURNAL OF THERAPEUTICS DOI10.1097/MJT.0000000000000916, 2019”).

- It would be useful to add a characterization of the risk profile of these patients, as well as a brief discussion regarding associated pathologies that may influence the evolution or mask the diagnosis - diabetes, hypertension, dyslipidemia, atrial fibrillation, smoking, etc.

- In the Discussion chapter, it should be specified that the comparative analysis with other studies is made subject to the fact that these patients are diagnosed late, therefore with a predictably more difficult evolution.

- In the Conclusions, it would be worth including a few brief comments on the importance of rapid recognition of critical lower limb ischemia, as well as possible suggestions for resolution strategies (modifications of the circuits of these patients, increasing awareness of this pathology and addressing the doctor as quickly as possible, etc.).

With these additions, I believe that the article can be published.

Author Response

Comment 1: "Enriching the introductory notions regarding the current treatment of patients with acute peripheral ischemia (see the article “New Directions in the Management of Peripheral Artery Disease Chioncel, V; Brezeanu, R; Sinescu, C AMERICAN JOURNAL OF THERAPEUTICS DOI10.1097/MJT.0000000000000916, 2019”)"

Response 1: Thank you for your comment. We have expanded the introduction and we have included the proposed reference, along with other relevant references (4 additional references have been added in the Introduction).

Comment 2: It would be useful to add a characterization of the risk profile of these patients, as well as a brief discussion regarding associated pathologies that may influence the evolution or mask the diagnosis - diabetes, hypertension, dyslipidemia, atrial fibrillation, smoking, etc

Response 2: Typical profile of patients with ALI has been added in lines 256-261 of the discussion. Actually, this typical profile is usually missing from patients with a delayed diagnosis, which could contribute to the delayed diagnosis.

Comment 3: "In the Discussion chapter, it should be specified that the comparative analysis with other studies is made subject to the fact that these patients are diagnosed late, therefore with a predictably more difficult evolution".

Response 3: A relevant comment has been added in Lines 227-229 of the Discussion.

Comment 4: "In the Conclusions, it would be worth including a few brief comments on the importance of rapid recognition of critical lower limb ischemia, as well as possible suggestions for resolution strategies (modifications of the circuits of these patients, increasing awareness of this pathology and addressing the doctor as quickly as possible, etc.)".

Response 4: A relevant comment has been added in the Conclusions, in Lines 331-334.

Reviewer 3 Report

Comments and Suggestions for Authors

Reviewer comments are listed below:

*Please lengthen the abstract to offer appropriate backdrop to the current study.

*Please separate "Embolism and Thrombosis" into two separate Keywords: Embolism; Thrombosis

*The introduction is far too scant to sufficiently provide the scientific premise of the study.  Please include items such as the disastrous outcomes of misdiagnosing and delaying ALI, why vascular surgeons are not the first specialists to potentially evaluate ALI, etc.

*The Materials and Methods are far too inadequate for any type of rigor and reproducibility to occur.  Did evaluations happen in a single hospital or multiple institutions?  The patient demographics is also insufficient.  What is the sex/gender, medications, race/ethnicity, exclusion/inclusion criteria, etc.?  Due to the low amount of data added to this manuscript, all of this information should be included in this study to ultimately enrich it.  The IRB information also appears inadequate in its current form.

*For the Results, how exactly was "misdiagnosis" determined?  Please clarify.  Also, was the appropriate diagnosis always from a vascular surgeon or any clinician?  This should be explicit.

*The Discussion section needs a complete overhaul be placing into context how the authors' data may contribute to the scientific community, how their data supports or refutes previous data, future directions, etc.  With this manuscript only currently having 10 citations, many more references need to be cited to enrich the Discussion.

*Please combine the Discussion & Conclusions sections.

*Please omit the "Patents" title.

Author Response

Comment 1: "Please lengthen the abstract to offer appropriate backdrop to the current study".

Response 1: We have lengthen the abstract according to this observation (Lines 13-16).

*Comment 2: "Please separate "Embolism and Thrombosis" into two separate Keywords: Embolism; Thrombosis".

Response 2: This has been modified accordingly.

Comment 3: "The introduction is far too scant to sufficiently provide the scientific premise of the study.  Please include items such as the disastrous outcomes of misdiagnosing and delaying ALI, why vascular surgeons are not the first specialists to potentially evaluate ALI, etc".

Response 3: We have expanded the Introduction according to this comment, by adding 4 references and a new paragraph (Lines 38-49 and Lines 53-55). 

Comment 4: "The Materials and Methods are far too inadequate for any type of rigor and reproducibility to occur.  Did evaluations happen in a single hospital or multiple institutions?  The patient demographics is also insufficient.  What is the sex/gender, medications, race/ethnicity, exclusion/inclusion criteria, etc.?  Due to the low amount of data added to this manuscript, all of this information should be included in this study to ultimately enrich it.  The IRB information also appears inadequate in its current form".

Response 4: Relevant information has been added in the Materials and Methods section, in Lines 75-84 and Lines 97-104. Data regarding dempgraphic information of participant, has been added in the 1st paragraph of the Results section.

Comment 5: "For the Results, how exactly was "misdiagnosis" determined?  Please clarify.  Also, was the appropriate diagnosis always from a vascular surgeon or any clinician?  This should be explicit".

Response 5: The definition for misdiagnosis that has been used in the current study has been reported in the Materials and Methods section, in Lines 73-76. The definitive diagnosis was made by vascula surgeons, as reported in Lines 77-79. 

Comment 6: "The Discussion section needs a complete overhaul be placing into context how the authors' data may contribute to the scientific community, how their data supports or refutes previous data, future directions, etc.  With this manuscript only currently having 10 citations, many more references need to be cited to enrich the Discussion".

Response 6: The Discussion has been expanded according to this comment. In total, 11 additional references have been cited in the revised manuscript. In Lines 303-318 detailed suggestions on how to improve rates of a timely diagnosis of ALI have been included.

Comment 7: "Please combine the Discussion & Conclusions sections".

Response 7: This has been modified accordingly.

Comment 8: "Please omit the "Patents" title".

Response 8: This has been omitted.

Reviewer 4 Report

Comments and Suggestions for Authors

The manuscript addresses an important clinical issue of misdiagnosis of acute limb ischemia (ALI) by non-vascular specialists and its impact on patient outcomes.

I have several major concerns regarding the manuscript:

1.       The analysed patient group is small and highly diverse in terms of age, causes of ALI, medical procedures performed, length of hospitalization, and other factors. This is not analyzed in greater depth. Why?

2.       Why were the cases of correctly diagnosed individuals not analyzed? How did their cases differ from those of misdiagnosed individuals? What were the characteristics of these patients?

3.       The manuscript does not include any statistical analyses to validate the observed differences or outcomes.

4.       Broader comparative analyses, particularly in the context of correctly diagnosed patients within the analyzed hospital, should be conducted. Similarly, comparisons with external datasets or studies could provide a more comprehensive understanding of the findings.

5.       The conclusions of the conducted analyses are unclear. How is this analysis supposed to assist in improving the diagnosis of patients with ALI? What guidelines for specialists performing the initial evaluation of the patient emerge from this analysis that have not been previously indicated in other studies?

Author Response

Comment 1: "The analysed patient group is small and highly diverse in terms of age, causes of ALI, medical procedures performed, length of hospitalization, and other factors. This is not analyzed in greater depth. Why?".

Response 1: Relevant comments have been included in the Discussion section, in Lines 251-264.

Comment 2: "Why were the cases of correctly diagnosed individuals not analyzed? How did their cases differ from those of misdiagnosed individuals? What were the characteristics of these patients?".

Response 2: The characteristics of patients with a timely diagnosis of ALI have been added in the 1st paragraph of the Results.

Comment 3: "The manuscript does not include any statistical analyses to validate the observed differences or outcomes".

Response 3:  Statistical comparisons between patients with a delayed vs those with a timely diagnosis, have been reported in the Results section, Subsection 3.4.

Comment 4: "Broader comparative analyses, particularly in the context of correctly diagnosed patients within the analyzed hospital, should be conducted. Similarly, comparisons with external datasets or studies could provide a more comprehensive understanding of the findings".

Response 4: Comparative analysis has been included in the revised version of our manuscript. Comparisons with previous studies have been added in Lines 304-313 (additional Refs that have been included #19-20), although available data are limited.

Comment 5: "The conclusions of the conducted analyses are unclear. How is this analysis supposed to assist in improving the diagnosis of patients with ALI? What guidelines for specialists performing the initial evaluation of the patient emerge from this analysis that have not been previously indicated in other studies".

Response 5: Additional comments have been included at the end of the Conclusion to adress this comment. Moreover a relevant paragraph including suggestions on how to improve the diagnostic pathway of these patients have been added in the Discussion, Lines 304-319.

Reviewer 5 Report

Comments and Suggestions for Authors

The manuscript entitled “Misdiagnosis of acute limb ischemia from non-vascular specialists results in a delayed presentation and negatively affects patients outcomes” is interesting and of great clinical value.

I am asking the authors to complete the Materials and Methods sections by describing of the clinical symptoms with which patients with a wrong diagnosis reported to doctors of various specialties. Moreover, there is no information on how the diagnosis was later confirmed; Doppler ultrasound or CT scan? The discussion lacks a description of typical symptoms of acute limb ischemia, what differential diagnosis should be performed, whether the symptoms should be differentiated from e.g. neuropathic pain, etc. More references are needed.

Author Response

Comment 1: "I am asking the authors to complete the Materials and Methods sections by describing of the clinical symptoms with which patients with a wrong diagnosis reported to doctors of various specialties". Moreover, there is no information on how the diagnosis was later confirmed; Doppler ultrasound or CT scan?".

Response 1: Information about the clinical presentation of patients has been added in Lines 108-112 of the Results. Moreover, information about the evaluation of patients during definitive diagnosis from vascular surgeons has been added in Lines 77-83 of the Materials and Methods.

Comment 2: "The discussion lacks a description of typical symptoms of acute limb ischemia, what differential diagnosis should be performed, whether the symptoms should be differentiated from e.g. neuropathic pain, etc".

Response 2: Relevant information regarding typical symptoms and differential diagnosis has been added in Lines 270-281 of the Discussion.

Comment 3: "More references are needed".

Response 3: Eleven additional references have been added in this revised version of the manuscript, for a total of 21 references in the current form of the submission.

Round 2

Reviewer 3 Report

Comments and Suggestions for Authors

The authors have tried a weak attempt at appeasing reviewer comments and their initial responses have done very little (if anything) to substantially strengthen the manuscript.  With this in mind, I encourage the authors at revisit the initial reviewer comments to adequately address them, keeping in mind, rigor, reproducibility, and study robustness.  If this cannot be accomplished appropriately, a stronger experimental design with higher quality methodology would be needed (a.k.a. de novo).

Comments on the Quality of English Language

English needs improvement.

Author Response

2nd Review, comment 1:

The authors have tried a weak attempt at appeasing reviewer comments and their initial responses have done very little (if anything) to substantially strengthen the manuscript.  With this in mind, I encourage the authors at revisit the initial reviewer comments to adequately address them, keeping in mind, rigor, reproducibility, and study robustness.  If this cannot be accomplished appropriately, a stronger experimental design with higher quality methodology would be needed (a.k.a. de novo)

Response: We would like to thank the reviewer for assessing our manuscript. Unfortunately, this review does not include any specific comments/observations. On the contrary, the reviewer suggests to address the initial comments that he had indicated at the 1st review. Although we have attempted to address all the reviewers remarks and we have modified our manuscript accordingly during the 1st revision, and despite the fact that in this 2nd revision no specific recommendations have been made except for a general comment, we have revisited the comments of the 1st review and attempted to add some further modifications and clarifications.

Regarding the conclusion of the reviewer, that “If this cannot be accomplished appropriately, a stronger experimental design with higher quality methodology would be needed (a.k.a. de novo)”, we would like to stress that this is NOT an experimental study, i.e. a randomized study. This is a retrospective observational study and as such, an experimental methodology is not applicable. Observational studies are an established type of studies in the literature. Of course they have limitations, and we have already included a relevant, Limitations paragraph at the end of our manuscript.

There follow the initial comments of the 1st review and some further answers/clarifications from the authors.

1st Review:

Comment 1: "Please lengthen the abstract to offer appropriate backdrop to the current study".

Response 1: We have further lengthened the abstract according to this observation (Lines 22-23 and 29-33).

*Comment 2: "Please separate "Embolism and Thrombosis" into two separate Keywords: Embolism; Thrombosis".

Response 2: This had been performed in the 1st revision.

Comment 3: "The introduction is far too scant to sufficiently provide the scientific premise of the study.  Please include items such as the disastrous outcomes of misdiagnosing and delaying ALI, why vascular surgeons are not the first specialists to potentially evaluate ALI, etc".

Response 3: During the 1st revision we had lengthened the Introduction and added 4 references. Now we have added lines 61-70 to address this comment, where it is reported that “Indeed, anecdotal experience suggests that it is common for patients with lower limb pain and related symptoms to be initially evaluated by other medical specialties such as general practitioners, orthopedics etc, which is especially true for patients without the typical atherosclerotic risk factors. Unfortunately, although this seems a reasonable argument, there are no published data to support such a claim. Moreover, the acute presentation of this condition, along with the limited resistance of neurologic and muscular tissue to ischemia and the significant risk of limb loss that previous research has indicated after ALI, render timely diagnosis and treatment critical for a favorable patient outcome”, which we believe answers to your comment.

Comment 4: "The Materials and Methods are far too inadequate for any type of rigor and reproducibility to occur.  Did evaluations happen in a single hospital or multiple institutions?  The patient demographics is also insufficient.  What is the sex/gender, medications, race/ethnicity, exclusion/inclusion criteria, etc.?  Due to the low amount of data added to this manuscript, all of this information should be included in this study to ultimately enrich it.  The IRB information also appears inadequate in its current form".

Response 4: We have attempted to further clarify this issue by adding Lines 91-96 in the Material and Methods section, where it is reported: “Patients that were initially evaluated from other medical specialties, were assigned a diagnosis different than ALI and were diagnosed with ALI >24h after initial presentation, were included in the analysis and comprised the study group. Patients with a timely diagnosis and standard treatment comprised the control group. All patients that were mis-diagnosed were initially examined in other institutions or in private practice, by non-vascular specialties. All patients were managed in a single institution. Initial or definitive diagnosis of ALI was made by vascular surgeons at the emergency department of the institution conducting this study, which provides the only vascular service in a large insular area in Greece. Therapeutic management and interventional/surgical procedures were performed at the same institution”.

Moreover, demographic information of the study population is reported in the 1st paragraph of the Results: “……There were  205 male and 75 female patients with a median age of 72years (33years-101years)…… Fourteen patients (5% of the total population) fulfilled the above-mentioned criteria of misdiagnosed ALI and were included in the analysis. Among them, there were 8 women and 6 men, with a mean age of 56years old (33 yo - 91yo)….. Most patients (243/280, n= 87%) were receiving either anticoagulant or antiplatelet treatment….”.

Comment 5: "For the Results, how exactly was "misdiagnosis" determined?  Please clarify.  Also, was the appropriate diagnosis always from a vascular surgeon or any clinician?  This should be explicit".

Response 5: Please see previous response where “misdiagnosis” is defined and that it is stated that definitive diagnosis was always made by vascular surgeon in the institution conducting the study.

Comment 6: "The Discussion section needs a complete overhaul be placing into context how the authors' data may contribute to the scientific community, how their data supports or refutes previous data, future directions, etc.  With this manuscript only currently having 10 citations, many more references need to be cited to enrich the Discussion".

Response 6: The Discussion had been modified during the 1st revision. 11 additional references had been added. 1st-4th paragraph discusses how the current results compare to the data of previous studies. 5th and 6th paragraph discusses how the atypical presentation of patients with ALI and the fact that many of them do not present the standard atherosclerotic profile of subjects with peripheral arterial disease, could have contributed in the delayed diagnosis. The 7th paragraph summarizes the typical symptoms and the differential diagnosis of ALI, which the non-vascular specialist should have in mind in order to correctly identify atypical presentation. In Lines 320-335, in the paragraph beginning with: “Our results indicate that there is certainly room for improvement on the diagnostic and therapeutic pathway of patients with ALI” future directions regarding improvement on the management of these patients are listed.

Comment 7: "Please combine the Discussion & Conclusions sections".

Response 7: This had been performed during the 1st revision.

Comment 8: "Please omit the "Patents" title".

Response 8: This had been performed during the 1st revision.

Reviewer 4 Report

Comments and Suggestions for Authors

The alterations performed have benefited the manuscript. I have no further remarks.

Author Response

Comment 1: The alterations performed have benefited the manuscript. I have no further remarks.

Response 1: Thank you for your comment and for reviewing this paper.

Reviewer 5 Report

Comments and Suggestions for Authors

Dear Authors, 

The manuscript is sufficiently improve.

Author Response

Comment 1: The manuscript is sufficiently improve.

Response 1:  Thank you for your comment and for reviewing this paper.